



# Future hot-spots for hydro-hazards in Great Britain: a probabilistic assessment

Lila Collet[1], Shaun Harrigan[2,3], Christel Prudhomme[2,3,4], Giuseppe Formetta[3], Lindsay Beevers [1]

[1]Heriot-Watt University, Edinburgh Campus, Edinburgh EH14 4AS, UK
[2]European Centre for Medium-Range Weather Forecasts, Shinfield Road, Reading, RG2 9AX, UK
[3]Centre for Ecology and Hydrology, Wallingford, OX10 8BB, UK
[4]Loughborough University, Epinal Way, Loughborough, LE11 3TU, UK

*Correspondence to*: Lila Collet (collet.lila@gmail.com)

**Abstract.** Hydrological extremes, floods and droughts, cause significant economic damages and pose risks to lives
worldwide. In an increasing hydro-climatic risk context as a result of climate change, this work identifies future hot-spots
across Great Britain expected to be impacted by an increase in both floods and droughts. First, flood and drought hazards
were defined and selected in a consistent and parallel approach with a threshold method. Then, a nation-wide systematic and
robust statistical framework was developed to quantify changes in frequency, magnitude, and duration, and assess time of
year for both droughts and floods, and the uncertainty associated with climate model projections. This approach was applied
to a spatially-coherent statistical database of daily river flows (Future Flows Hydrology) across Great Britain to assess
changes between the baseline (1961-1990) and the 2080s (2069-2098). The results showed that hydro-hazard hot-spots are
likely to develop along the west coast of England and Wales and across northeast Scotland, mainly during the winter (floods)
and autumn (droughts) seasons, with a higher increase in drought hazard in terms of magnitude and duration. These results
suggest a need for adapting water management policies in light of climate change impact, not only on the magnitude, but
also on the timing of hydro-hazard events, and future policy should account for both extremes together, alongside their
potential future evolution. This novel, consistent, method is transferable to new hydro-climatic projection databases.

## 1 Introduction

Hydrological extremes, floods and droughts, cause significant economic damages and pose risks to lives worldwide
(Quesada-Montana et al., 2018). In the UK, the government has estimated that annual flood damages of £1.1 billion are
anticipated and maintaining the current levels of flood defence would cost to as much as £27 billion by 2080 (UK
Parliament, 2013). At the same time, the UK's vulnerability to drought hazard has reached the warning threshold for the
Water Exploitation Index that defines it as a water-stressed country (EEA, 2008), and the financial impact of the recent
2011/12 drought was £70-165M. These risks, alongside their likely exacerbation associated with the future climate, has been
recognised by the UK Government Water White Paper (HM Government, 2011), which highlights that 'drought conditions
are likely to be more common'. These concerns are reflected in the Environment Agency research priorities (Environment





Agency, 2014) where 'understanding of hydro-hazards and their impact on people' within a changing climate is an area of critical importance to the nation. More recently, the Committee on Climate Change identified flooding and water supply shortage as two of the UK's most important climate change risks (ASC, 2016), their future high magnitude risks estimated with high confidence, suggesting that more action is urgently needed to face these issues.

Hydrological hazards are influenced by climatic and hydrological factors (e.g. rainfall patterns and intensity, land use, soil and bedrock etc.); accounting for their potential future changes into new development is hence essential to design resilient cities and their supporting infrastructure (Bai et al., 2018). However, detecting changes in observed records is complex. For example, observed records show increases in extreme precipitation over the past 50-60 years across the UK (Maraun et al., 2008), and in high river flows in western Britain (Hannaford and Marsh, 2008; Harrigan et al., 2017), but no substantial

changes were found for flood magnitude (Hannaford and Marsh, 2008). In parallel, potential evapotranspiration has increased in all regions of Great Britain between 1961-2012, mainly driven by rising air temperature, with strongest increases in spring and for England (Robinson et al., 2017). Rainfall intensity has increased in the winter and to a lesser extent during spring and autumn, while summer intensities have reduced. Historic precipitation records in the UK show diverging seasonal trends (increasing winter and decreasing summer precipitations, see Burt et al., 2016), and later winter

storms across the North Sea (Blöschl et al., 2017). Trends in extreme river flow (frequency and magnitude) have strong regional and geographical patterns, with low-flow magnitude between 1963-2014 showing a prominent spatial gradient with increases in the northwest and decreases in the southeast (Harrigan et al., 2017). Whilst trends are not always statistically significant everywhere, these changing patterns make future water management decisions difficult. .

Evidence of trends in the past hydro-climatic records suggests a non-stationary regime. This means that using historic

records is unlikely to be sufficiently robust when planning water resource management several decades ahead. Future planning should consider the possible evolution of the climate when estimating future hydro-hazards. Climate models are tools designed to provide scenarios of possible future precipitation and temperature patterns, which can be used to drive hydrological models and understand potential evolution of future hydro-hazards (Augustin et al., 2008; Arnell and Gosling, 2016; Collet et al., 2017). Studies suggest that climate change is expected to increase return period flow magnitude (e.g. Kay

et al., 2014a; 2014b; Collet et al., 2017; Kundzewicz et al., 2017; Collet et al., 2018), but there is significant uncertainty associated with these projections due to the uncertainties in the climate signal and the impact modelling chain (Kundzewicz et al., 2018). Drought patterns are also expected to be impacted, for example due to projected increases in dry spells and potential evapotranspiration (Trenberth, 2011; Fischer et al., 2013). Future changes in meteorological drought (Rahiz and New, 2013) and hydrological drought (Prudhomme et al., 2012a) in Great Britain show a mixed pattern, with increases found

across the country in the summer but largest in the north and west.

In the UK, most regions suffer from both floods and droughts, and can even be impacted simultaneously (e.g. 2010-2012 hydrological transformation in Southern UK, see Parry et al., 2013). Recent work on changes in observed floods and droughts using different approaches (e.g. the return-period method across the UK in Burt et al., 2016 and the threshold level approach on one catchment at the monthly time step in Quesada-Montano et al., 2018) show a growing need and interest in



understanding changes in hydrological dynamics across the full flow regime. Moreover, understanding the possible future evolution of both hydro-hazards is critical for building resilient solutions to climate change. This is particularly important for regions expected to become even more at risk of both floods and droughts, as these would be 'hot-spots' where resilience to hydro-hazards must be strengthened. However, generally floods and droughts are considered independently in water

management planning. To our knowledge there is no analysis to date investigating possible future changes in the frequency, magnitude, and duration of both hazards in Great Britain using a consistent methodology; nor investigating whether increases in both floods and droughts are expected in the same part of the country or whether the hazards are geographically distinct.

This work aims to identify future hot-spots across Great Britain expected to be impacted by an increase in both floods and

droughts. We develop and apply a nation-wide systematic, consistent and robust statistical framework to quantify changes in frequency, magnitude, duration, and time of year of both drought and flood, and their associated uncertainty.

## 2 Data and Methods

### 2.1 The Future Flows Hydrology dataset

The Future Flows Hydrology (FFH) database (Prudhomme et al., 2013) is currently the only nation-wide, consistent,

probabilistic future transient hydrological projection available for the UK. Future Flows Hydrology is derived from the Future Flows Climate (Prudhomme et al., 2012b), a national, 11-member ensemble projection derived from the UK Met Office Hadley Centre's ensemble projection HadRM3-PPE. HadRM3-PPE-UK was developed as part of the derivation of the UKCP09 scenarios (Murphy et al., 2007) and designed to represent parameter uncertainty in climate change projections through a parameter variant experiment and was run under the SRES A1B emissions scenario (see Murphy et al., 2009

which details the climate model perturbations). Future Flows Climate was used as forcing for three hydrological models (CERF, see Griffiths et al., 2006, PDM, see Moore, 2007, and CLASSIC, see Crooks & Naden, 2007) to create the Future Flows Hydrology database, which contains an 11-member ensemble of transient projections of daily river flow for 281 catchments from January 1951 to December 2098. Each FFH member is associated with a single realisation from a different variant of HadRM3, each member representing an equally probable, plausible realisation of the future (Murphy et al., 2007).

### 2.2 Hydro-hazard analytical framework: event extractions

Each daily river flow series was analysed across the 11 ensemble-members to detect changes in high and low flows between two time periods: the baseline (1961-1990) and the 2080s (2069-2098). A threshold-based method was applied to both flood and drought hazards, to ensure consistency and comparability of results (see Fig. 1 and Table 1). For floods we used the peak over the threshold (POT) series (Stedinger el al., 1993; Robson & Reed, 1999) and for droughts its equivalent, the Inter-

event time and volume Criterion (IC) method (see e.g. Gustard & Demuth, 2009). High- and low-flow thresholds were





defined to obtain on average three independent events per year on the baseline period, with the same threshold applied for the 2080s period.

Flood characteristics were analysed following the Peak-Over-Threshold method of Bayliss and Jones (1993). Here, each ensemble member discharge simulation was treated independently, with a threshold selected for each member so that an average 3 independent flood events per year could be identified during the baseline period, a flood event being the period when the daily discharge curve is continuously above the threshold (dashed line in Fig. 1.a, see example for high-flow event number 1). The mean number of 3 POTs per year has been fixed to compute the threshold in the baseline period, and the same threshold is used in the 2080s, hence the mean number of independents events in the 2080s could change. For each independent flood event, peak magnitude (highest daily discharge within the period), duration (number of days of the event) and date of highest peak (high-flow event number 2 in Fig. 1.a) were extracted.

Drought characteristics were analysed following the method from Gustard & Demuth (2009) using the R package 'lfstat' (available at https://cran.r-project.org/web/packages/lfstat/index.html). As for floods, each ensemble member was treated independently. Here, after a sensitivity analysis on drought event frequency, a daily varying Q90 threshold (i.e. the flow which was equalled or exceeded 90% of the time over each Julian day across the 30-year baseline) was applied to select on average 3 independent low-flow events per year on the baseline (see grey line in Figure 1.b). Dependent events were pooled together applying the IC method (Gustard & Demuth, 2009) using a minimum of 5 days inter-event time, and a 0.1 ratio between inter-event excess volume and preceding deficit volume. For each pooled low-flow event, magnitude (water volume deficit, i.e. the amount of water between the daily Q90 threshold and the daily discharge; see grey areas), duration (number of days the daily discharge curve is below the daily Q90 curve; see low-flow event number 2) and dates (date of the minimum discharge during a low-flow event, see low-flow event number 3) were extracted (see Fig. 1.b and Table 1). Since the threshold used to detect low-flows varies at a daily time-step, both summer and winter events were selected. This supports the need to understand water volume deficit across the year to comprehend drought risk. Indeed summer water deficits clearly became stronger in the twentieth century in Great Britain as a result of increasing temperatures mainly, although winter rainfall – and potentially winter flows – influences groundwater recharge and reservoir supply (Marsh et al., 2007; Fowler & Kilsby, 2012).

Finally, summary characteristics for each flood and drought series were calculated for both baseline and 2080s period (see Table 1): frequency, as the mean number of independent events per year; magnitude as the mean annual maximum POT (floods) and annual cumulative water deficit (droughts); duration as the mean annual cumulative duration of all events. In addition, two time of year metrics were extracted from dates of max/min flows using circular statistics following the approach of Bayliss and Jones (1993) and Institute of Hydrology (1999), i) the 'mean day of year' of events and ii) the concentration of dates around the mean day of year, known as 'seasonality' with values ranging between 0 for when floods/droughts are widely dispersed throughout the year (no concentration) and 1 if floods/droughts occur on the same day each year in the record (see e.g. Formetta et al., 2018).





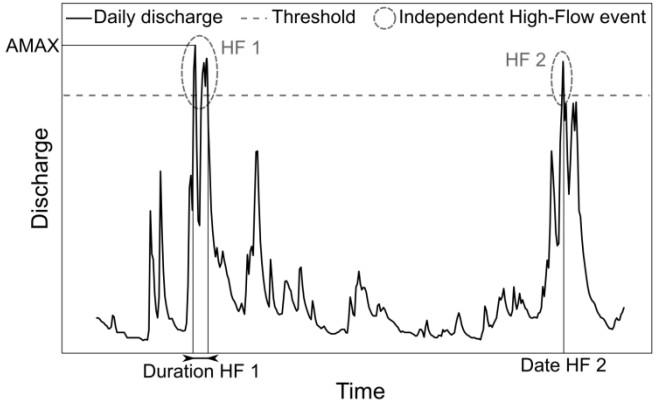 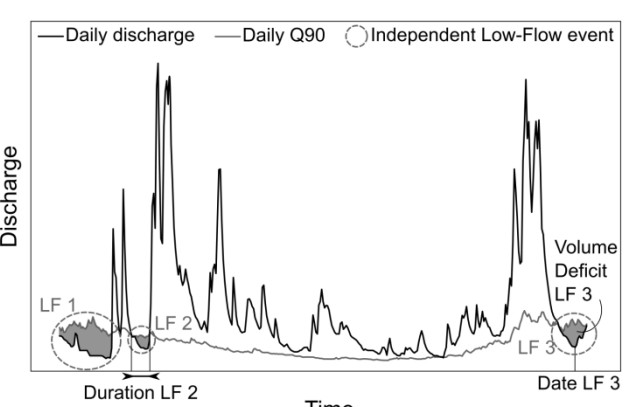

**Figure 1: Selection and characterisation in terms of frequency, magnitude, duration, and time of year of: (a) Floods (HF: High-Flow event, AMAX: Annual Maximum POT), and (b) Droughts (LF: Low-Flow event).**

**Table 1: Summary of flood and drought characteristics.**

| Characteristic | Floods | Droughts |
|---|---|---|
| Magnitude | Annual maximum POT | Annual volume deficit |
| Frequency | Number of independent peaks over threshold | Number of pooled low-flow events under threshold |
| Duration | Number of days over threshold | Number of days under threshold |
| Time of year | Date of maximum peak flow | Date of minimum flow |

### 2.3 Hydro-hazard hot-spot assessment

Hydro-hazard hot-spots were selected based on changes in flood and drought characteristics from the baseline to the 2080s. First, the frequency, magnitude, duration, and time of year were computed for each ensemble-member for both the baseline and 2080s periods. Change in floods and droughts in terms of frequency, magnitude, and duration were quantified as the differences between the baseline and 2080s values, which were computed for the 10[th], 50[th], and 90[th] percentiles of the 11-member distributions (Fig. 2). Uncertainty in the signal of change was quantified as the range of changes computed across the three investigated percentiles.





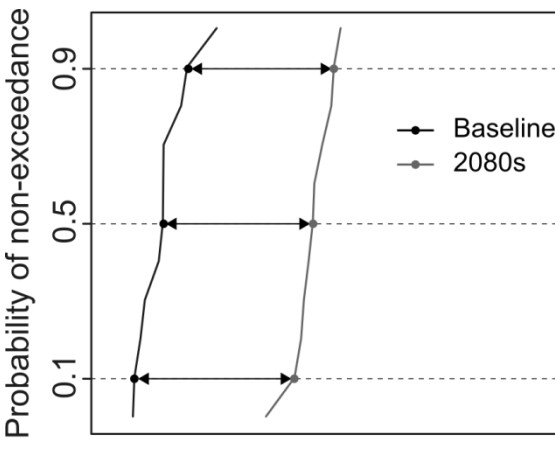

**Figure 2: Quantification of signal of change in a hydro-hazard characteristic, as the difference between the baseline and 2080s cumulative distribution functions for the 10th, 50th, and 90th percentile (dotted lines).**

Next, hot-spots were identified across the UK based on prominent changes in flood and drought characteristics for each of

5  the 10th, 50th, and 90th percentiles (see Fig. 3). A catchment was defined as a hot-spot if, for both floods and droughts, it showed an increase in:

-      frequency (above +1day/year, see Fig. 3a),
-      AND in magnitude (above 5%, see Fig. 3b),
-      AND in duration (above +1 day/year for floods, above +5 days/year for droughts, see Fig. 3c).

10  These thresholds were chosen after a sensitivity analysis (not shown here) to find an acceptable amount of catchments for each percentile. Note that in terms of duration, since floods are by nature shorter events, an increase in duration by 1 day/year was found to be a reasonable discriminator whereas a larger increase in duration of drought events was necessary to characterise changes in these events. The resulting catchments were mapped for each percentile and the changes in each characteristics were analysed spatially.

15        Finally, the time of year (i.e. the mean day of year and seasonality) of these events in the 2080s were mapped for the hot-spot catchments, for each percentile. This shows the month when these hydro-hazards would happen in order to analyse how the hydrological regime would change with climate change, i.e. characterizing when these extremes would intensify across the year.





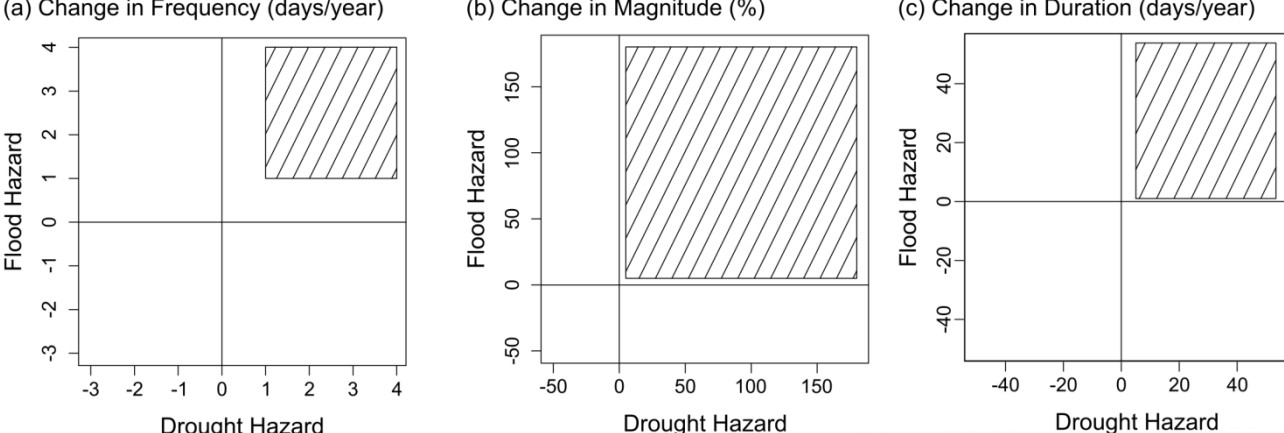

**Figure 3: Selection of hot-spots based on changes in (a) frequency, (b) magnitude, and (c) duration of both flood and drought hazards (dashed areas).**

## 3 Results

### 3.1 Hydro-hazard hot-spots

Figure 4 shows the catchments identified as hydro-hazard hot-spots in Great Britain for the $10^{th}$ (Fig. 4a), $50^{th}$ (Fig. 4b), and $90^{th}$ (Fig. 4c) percentiles across the 11 ensemble-members of the Future Flows Hydrology database. Only two catchments were identified for the $10^{th}$ percentile (Fig. 4a) in Wales (Gwili River at Glangwili) and Scotland (Ruchill Water at Cultybraggan). For the $50^{th}$ percentile, representing the median trend across the 11 climatic projection ensemble-members, 48 catchments are defined as hot spots (Fig. 4b), mainly located on the west coast and northeast of Scotland. The $90^{th}$ percentile shows 135 catchments (Fig. 4c) spread throughout Great Britain. These hot-spots are the result of a combination of changes in droughts and floods characteristics, which are detailed in the following sections.




◆ Gauging Station       ● Hydro-Hazard Hot-Spot      0   100   200 km  N

(a) 10<sup>th</sup> Percentile      (b) 50<sup>th</sup> Percentile      (c) 90<sup>th</sup> Percentile

**Figure 4: Hydro-hazard hot-spots in Great Britain for the (a) 10th, (b) 50th, and (c) 90th percentiles.**

**3.2 Changes in frequency**

Figure 5 shows the changes in frequency of the hydro-hazard for the hot-spots across Great Britain depicted in Fig. 4. For the
10th percentile, the two hot-spot sites show increases to the frequency of floods and droughts of between 1 and 2 events per
year by the 2080s (see Fig. 5a). For the 50th percentile, the majority of the 48 identified hot-spots show an increase in
frequency of floods and droughts of 1-2 events per year (see Fig. 5b). Three sites (two in the southwest of England, and one
in Wales) show an increasing frequency of 2-3 events per year for floods only and one site in Wales shows the same increase
for droughts only. For the 90th percentile, sites in the southwest of England, Wales, and the northeast of Scotland show a
greater increase (2-3 events per year) for flood events than for drought (see Fig. 5c). Three sites in the southwest of England
and one in the southwest of Scotland suggest an increase in frequency by 3-4 events per year for floods; while droughts at
the same location increase in frequency by 1-2 events per year. The spatial distribution of increasing frequency of droughts is
different, with increases (2-3 events per year) notable across the central belt in Scotland, central England, and Wales, whilst





flood increases are generally lower at 1-2 events per year. In general there is reasonable agreement across the ensemble members for the hot-spots; suggesting constrained uncertainty in frequency increases.



**Figure 5: Changes in frequency of the hydro-hazards for the hot-spots in Great Britain for the (a) 10<sup>th</sup>, (b) 50<sup>th</sup>, and (c) 90<sup>th</sup> percentiles.**

### 3.3 Changes in magnitude

Figure 6 shows the changes in hydro-hazard magnitude for the identified hot-spots across Great Britain. For the 10<sup>th</sup> percentile, both hot-spot sites show contrasting results (see Fig. 6a). For the Scottish site both floods and droughts are likely to experience an increase in magnitude of between 5-20%, while for the Welsh site floods increase by a much lower magnitude (5-20%) than droughts (100-150%). For the 50<sup>th</sup> percentile, a clear trend in more severe droughts is emerging (see Fig. 6b). The hot-spot sites suggest an increase in drought magnitude in southwest and northwest England, Wales, and northeast Scotland of between 50-150%; whilst flood magnitude increases are significantly less with the majority of sites, increasing by 5-20% and only 11 sites showing an increase in magnitude of 20-50%. For the 90<sup>th</sup> percentile (Fig. 6c), all hot-


spot sites suggest an increasing drought magnitude above 20% (with the exception of four stations in the south of England). Drought magnitude increases are most notable in the west of Great Britain, across the central belt, and in the northeast of Scotland. Flood magnitudes are more constrained with all hot-spot sites suggesting an increase in magnitude of below 50%. These results suggest that the increase to hydro-hazard magnitude may be more strongly evident in droughts in the future; although the uncertainty associated with this projection is higher (greater range in results at each station) for droughts than for floods.

**Figure 6: Changes in magnitude of the hydro-hazards for the hot-spots in Great Britain for the (a) 10th, (b) 50th, and (c) 90th percentiles.**

### 3.4 Changes in duration

Figure 7 shows the changes in duration of the hydro-hazards for the hot-spots across Great Britain. Due to the nature of drought (i.e. its longer temporal signature) the level at which changes were screened was +5 days for drought and +1day for



floods (see section 2.3). With that in mind, for the 10th percentile (Fig. 7a), both hot-spots suggest an increase in flood duration of 1-5 days per year, and an increase in droughts of 5-30 days per year. For the 50th percentile (Fig. 7b), the majority of the 48 identified hot-spots suggest an increase in droughts of between 5-30 days per year. Nine stations suggest a higher increase of between 30-55 days. These stations are located in southwest England (7), Wales (1) and northeast

5    Scotland (1). All but two stations suggest an increase in flood duration of between 1-5 days per year. One station on the south coast and one in Wales suggest a more severe increase in flood duration of between 5-30 days per year. For the 90th percentile (Fig. 7c), increases to drought duration are split between 5-30 days and 30-55 days. The more severe increases in duration are experienced in northeast Scotland, northeast England, through central England, Wales, and the southwest. Increases in flood duration remain predominantly between 1-5 days per year. Only a few stations suggest an increase above 5

10    days per year, and these are located in the southwest of England, peninsular Wales, one in central England, and one in northeast Scotland. In general the increase in duration of flood events is much more constrained than for droughts. This is partly due to the longer temporal signature of drought phenomena, rather than floods; but is also due to the fact that Great Britain is an island, with small catchments and relatively short flood events.



**Figure 7: Changes in duration of the hydro-hazards for the hot-spots in Great Britain for the (a) 10th, (b) 50th, and (c) 90th percentiles.**

### 3.5 Time of year of hydro-hazards in the 2080s

Figure 8 shows the hydro-hazards time of year for the identified hot-spots across Great Britain in the 2080s. For the 10th
percentile (Fig. 8a), the mean day of year of floods falls in early winter (December) while droughts occur in early
(September for the Scottish hot-spot) and late (November for the Welsh hot-spot) autumn. In the 2080s, the seasonality is
much stronger for floods (0.4-0.6) than for droughts (below 0.4), suggesting flood events would more consistently occur in
winter-time and the droughts mean day of year is not significant for this percentile. For the 50th percentile (Fig. 8b), the
majority of catchments show flood events occurring in winter-time (December or January) while droughts occur in autumn.
Only eight hot-spots in northern Wales, northern England, and Scotland show floods and droughts both in autumn and one
site in northwest of England shows drought events in early spring. This shows mean day of year are more consistent for
floods than for droughts in the 2080s since the seasonality shows higher values for the former than the latter. The seasonality
of these hot-spots is higher for floods (75% of hot-spots above 0.6) than droughts (94% of hot-spots below 0.4), showing
floods events more concentrated in the winter time while droughts would be more spread out across the year. For the 90th
percentile (Fig. 8c), Wales, England, and southwest and northeast of Scotland show winter floods coupled with autumn
droughts, while the north and central belt of Scotland show both floods and droughts in autumn. There is a national split with
earlier events in the northwest of the country (late autumn for floods and early autumn for drought) and later events in the
southeast of England (late winter for floods and late autumn for droughts). Once again the seasonality of these events is
higher for floods (69% of hot-spots above 0.6) than droughts (71% of hot-spots below 0.4). For the identified hot-spots, the
time of year is consistent across the ensemble members, showing a low uncertainty in this variable in the forcing signal from
the Regional Climate Model.





**Figure 8: Time of year (mean day of year in colour scale and seasonality in size scale) of the hydro-hazards for the hot-spots in Great Britain in the 2080s for the (a) 10ᵗʰ, (b) 50ᵗʰ, and (c) 90ᵗʰ percentiles.**

## 4 Discussion

### 4.1 Understanding hydro-hazard hot-spots

British hydro-hazard hot-spots are identified mainly along the west coast and in northern Scotland. Indeed results show a marked northwest-southeast gradient across Great Britain for changes in both droughts and floods according to the FFH database. The west coast shows smaller but more likely increases to flood hazard in the 2080s (in terms of frequency, magnitude and duration), and a higher increase to magnitude and duration in drought hazard from the baseline to the 2080s. In the baseline (see Fig. 9), the seasonality of droughts is very low (below 0.2 for all the stations of the 10ᵗʰ and 50ᵗʰ percentiles and below 0.4 for all the stations of the 90ᵗʰ percentile), showing that the mean day of year is not representative for these events, while for floods seasonality is high (above 0.6), showing these events occur mainly in late autumn (west



coast and northeast of Scotland) and winter on the baseline. In the 2080s, while floods would still occur mainly during the late autumn and winter season, drought events would be more concentrated in autumn, with a significantly higher seasonality. This shows a likely intensification of hydrological extremes in this part of Great Britain that would imply a need to adjust water management plans for both hydro-hazards.

**Figure 9: Time of year (mean day of year in colour scale and seasonality in size scale) of the hydro-hazards for the hot-spots in Great Britain on the baseline for the (a) 10th, (b) 50th, and (c) 90th percentiles.**

Increases in multi-day and extreme precipitations are expected as a result of climate change in the north and west of Great Britain (Wilby et al., 2008), which would translate into rising high-flow magnitude. Changes in 1:100-year return period events as a result of climate change showed a higher increase in the southeast of England (Collet et al., 2017), which is consistent with the spatial distribution of results in this study. These future changes would be the continuity of observed trends found in the literature. Harrigan et al. (2017) showed a significant increase in observed high-flows over 1965-2014 across near-natural catchments in the United Kingdom, particularly in Scotland, which is explained by wetter winter and



autumn seasons, and Blöschl et al. (2017) showed temporal shifts of observed floods to earlier winter season in Scotland and northern England.

The changes in low-flows are highly constrained by the Future Flows Climate (FFC) dataset, which was used to generate Future Flows Hydrology (Prudhomme et al., 2012b). Drought propagation from the meteorological to the hydrological signal

can show a fair linearity in temperate climates, such as the British oceanic climate, being mainly driven by precipitation and temperature patterns (Van Loon et al., 2014), particularly for catchments with low influence from groundwater dynamics. For the medium scenario (A1B), the UKCP09 projections in the 2080s result in winter precipitations that suggest a higher increase in the southeast of England than on the west coast; while future summer precipitations range from a significant decrease to a slight increase, with a wider uncertainty in the south and southeast of England. Rahiz and New (2013) analysed

changes in monthly precipitation series of the HadRM3-PPE-UK database, the same ensemble of regional climatic projections that were then downscaled to create the FFC dataset. Maps of Drought Intensity (DI) in the 2080s calculated based on the 6-month drought severity index show that the increase in hydrological drought found in their study for the west of Great Britain is mainly explained by an increase in DI in summer.

Through a consistent analysis of changes in both the high and low extremes in terms of frequency, magnitude, duration, and

time of year, this study brings new insights on plausible climate change impacts on hydro-hazards. This systematic approach across Great Britain highlights how both hazards evolve spatially in the future and quantifies the magnitude and temporal shifts of these changes. These outputs show a holistic overview of changes in hydrological seasonal variation. The statistical approach provides a direct insight into the uncertainty related to climatic projections and helps quantify the likelihood of such projections over the long-term. These insights are crucial to anticipate future climate change impacts on the

hydrological regime, and can help prepare improved adaptation plans in the context of increasing hydro-climatic risk. Consequently such analyses can inform water managers' future adaptation strategies and assist in anticipating new water infrastructure scheduling and timing.

## 4.2 Implications for Water Resources Management

Results of this study showed that changes in high-flow magnitude and duration would vary spatially across Great Britain.

This spatial distribution needs to be acknowledged by authorities for better flood risk management plans. Across Europe flood policy has generally adopted a risk based approach through the EU Floods Directive 2007/60/EC to deal with future changes in flood hazard. As part of the Directive, member states have prepared flood hazard maps and risk management plans in their region to anticipate changes in peak flows (Kundzewicz et al., 2012). In the UK, until 2016, climate change safety factors of 10-20% (by 2025 and 2080 respectively) were adopted across the country (Reynard et al., 2017), which

would underestimate the possible increase in high-flows for many catchments. In 2016 these factors were updated to reflect the regional influence of geographic, geological, and hydrological factors on the climate change response across England and Wales (Scotland and Northern Ireland are in the process of changing their guidance). This new guidance recognizes the





uncertainty in climate projections and subsequent responses by providing a range of uplift factors for different time periods and catchment regions (Kay et al., 2014a).

Moreover, changes in the hydrological cycle dynamics can lead to changes in the physical system and response of the river to meteorological events. An increase in frequency, as well as magnitude, of peak flows can significantly change the

morphology of the river channel, through sediment transport (Pender et al., 2016). Changing flow regimes influence sediment transport rates, erosion and depositional zones. These links mean that not only could more out of bank events occur, they may additionally trigger a change in river morphological response, resulting areas of deposition in constrained urban channels. In turn this may change the channel shape and hence result in chances to flood protection design being overtopped, as the morphological considerations of channel change tend not to be included in flood risk assessments.

Possible changes in both extreme flood and drought risks need to be investigated and monitored in local management plans to better anticipate future changes in the water cycle at the catchment scale.

Finally, as stated before, the time of year is strongly dependent on the chosen threshold. In this study the threshold calculated on the baseline was applied to the 2080s series. For high-flows, a constant threshold was selected, and no significant change in time of year was found from the baseline to the 2080s. However regarding low-flows, the baseline daily-varying threshold

suggests a certain seasonal variation of river flows, which can be accounted for in water management plans. According to the FFH database no particular time of the year is emphasised on the baseline. However, when compared to the baseline, results suggest that there would be more water deficit in autumn and winter in the 2080s. These trends could result in multi-seasonal drought (or "wet-to-dry-season drought", as defined by Van Loon and Van Lanen, 2012) if occurring after a significant summer low-flow, as recharge would not fully recover in winter as expected, which could also result in a more severe low-

flow in the following summer season. Indeed in England and Wales winter rainfall is key to groundwater recharge, which is the principal source of river flow in summer, showing these regions are particularly vulnerable to winter droughts (Marsh et al., 2007). These results suggest a need for adapting water management policies in light of climate change impact, not only on the magnitude, but also on the timing of low-flow events. This should be considered in the full context of hydro-hazards and water management where large infrastructure is part of the river basin. For example, reservoir rule curves which account

for flood management storage over winter may need to be revisited in order to assess the potential to manage for dual purposes.

## 4.3 Limits of the study

Some limits of this study are related to the use of the Future Flow database. As reported by Prudhomme et al. (2013), three hydrological models are used to simulate river flow with the emphasis of calibration on different parts of the flow regime.

The CERF model was calibrated mainly on the representation of the water balance and low flows, while for PDM and CLASSIC the emphasis is on the upper part of the flow regime and peak flows. For the gauging stations calibrated with the CERF model, the high-flows might thus be under-estimated while for the gauging stations calibrated with the PDM and CLASSIC models, the low-flows might be over-estimated.



Moreover, this study investigates the uncertainty related to one climate model only (HadRM3), under one forcing scenario (SRES A1B). The FFH database is based on a downscaled subset of the UKCP09 database, the HadRM3-PPE-UK, which does not capture the full range of the climate variable space projected by UKCP09 (Prudhomme & Williamson, 2013). For example, when using outputs from the UKCP09 weather generator (Murphy et al., 2009) with a range of different emission

scenarios, changes in peak flows show a different spatial distribution (higher increase of 1:20-year return period events in the west), with a wider uncertainty (Kay et al., 2014a; 2014b). Investigating hydrological data derived from climatic projections forced by a wider range of emission scenario would thus probably lead to a larger range of possible changes in high and low-flows (Wilby & Dessai, 2010). Finally, using outputs from diverse General Circulation Models would allow the inclusion of a wider set of possible futures in impact studies to assess the probability and uncertainties related to these models and

scenarios (Wilby, 2010). However, when comparing UKCP09 to the Coupled Model Inter-Comparison Project Phase 5 (CMIP5) projections, which were used in the Intergovernmental Panel on Climate Change 5th assessment (IPCC AR5) and reflecting on the uncertainties related to the GCM structure, the current recommendation is that UKCP09 provides consistent results for future changes to summer and winter temperature and winter rainfall (Met Office Hadley Centre, 2016). The main differences were found for future summer rainfall changes: while both experiments agree on a likely future reduction over

the long-term, CMIP5 suggests a smaller likelihood of substantial future reductions, especially for England and Wales.

## 5 Conclusion and Perspectives

In the context of increasing hydro-climatic risk arising from climate change, this paper aims to characterize the changes in flood and drought hazards spatially, temporally, and by magnitude in a consistent and parallel approach. It also embraces the uncertainties related to climatic projections and provides a framework to quantify the likelihood of these changes. A

systematic approach is thus developed and applied to a spatially-coherent statistical database of daily river flows (Future Flows Hydrology) across Great Britain to assess changes between the baseline (1961-1990) and the 2080s (2069-2098). This method characterizes changes in frequency, magnitude, duration, and time of year of flood and drought hazards consistently, and identifies future hotspots across the country. Results showed that the FFH projects hydro-hazard hot-spots along the west coast of England and Wales and in Scotland, mainly during the winter (floods) and autumn (droughts) seasons, with a higher

increase in drought hazard in terms of magnitude and duration. Some limits to this study relate to the ability of the hydrological models (used to produce the FFH database) to reproduce extreme high and low-flows, while others are associated with FFH's limitation to one single climate model and emission scenario (SRES A1B). However this paper sets out a novel approach to characterize both flood and drought hazards in a consistent manner across a large territory and in a probabilistic framework.

This paper presents a robust methodological approach to identify hydro-hazard hot-spots over a large spatial domain. The FFH database is a unique spatially coherent national-wide statistical river flow database, and presents an opportunity to develop methods to quantify climate change impacts (and its associated uncertainty) on hydrological extremes. It can be





transferred to other large-scale statistical hydrological products that are emerging, such as the End-to-end Demonstrator for improved decision-making in the water sector in Europe experiment (EDgE, http://edge.climate.copernicus.eu/), which shows a growing interest towards large-scale impacts of climate change on the hydrological cycle by stakeholders and a need for practical end-user available data on this matter. Moreover, the upcoming UKCP18 (http://ukclimateprojections.metoffice.gov.uk/24125) based on IPCC AR5 should also provide appropriate downscaled climatic projections for the UK, the development of these new climate scenarios being driven by both the climatic and the end-user communities. This work is not an attempt to present the most state-of-the-art climate change projection chain, but rather to develop a novel methodological approach to characterize changes in both hydrological extremes as a result of climate change. This method is thus now transferable to these upcoming new databases to understand climate change impact on hydro-hazards and ultimately inform stakeholders and decision-makers. The output maps can be used to select case studies and investigate changes in floods and droughts in a risk assessment framework. Cascading uncertainties into impact studies has been being investigated for vulnerability, resilience and risk assessment. However the literature shows this has been studied separately for flood (see e.g. Di Baldassarre et al., 2009; Masood & Takeuchi, 2012) and drought (see e.g. Borgomeo et al., 2015; Collet et al., 2015) risks. Building upon the work presented here, future risk assessments should consider both flood and drought hazards in a common and coherent framework.

## Author Contribution

LC and GF developed the codes and performed the data analysis. LC prepared the manuscript with contributions from all co-authors.

## Acknowledgements

This work was carried out as part of the EPSRC EP/NE30419/1 project 'Water Resilient Cities'. Acknowledgement is also given to the NERC-CEH Water Resources Science Area.

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
