# Peer review of "Future hot-spots for hydro-hazards in Great Britain: a probabilistic assessment"

_Hydrology and Earth System Sciences, 2018_

## Referee Comment (RC1) · Anonymous Referee #1 · 17 Jul 2018

The authors have provided a well-prepared manuscript that presents a technique for identifying sites experiencing concurrent increases in the frequency, magnitude and duration of both floods and droughts. The research is sound and the presentation is clear. With the addition of some points for discussion, I do not see a major impediment to the publication of this work. The following comments are provided to provoke discussion and support the authors to deepen the impact of this work.

My main concern is that the work seems to identify only locations with increasing magnitudes of hydrologic hazard, rather than hot-spots of hazards. While this is fine, it needs to be clearly stated. As in line 30 of page 17, there is a tendency to inadvertently slip into calling these locations "hydro-hazard hot-spots" rather than remaining "in the context of increasing hydro-climatic risk" (line 17, page 17). While the nuance

seems pedantic, I think it important to observe that a location might experience a high probability of hydrologic risk without that risk significantly increasing. Still further, it is possible that, while a location may not be categorized as a hot-spot here, it may still be experiencing stark increases in one or two of categories identified.

On a similar note, I found myself wondering about the locations that experienced some subset of the three criteria described on page 6 and shown in Figure 3. Using the strict criteria that all three be represented is useful, but I would like to see some note about sites that were missed and how strong the correlation between criteria is. (That is, if a site meets the Hazard criteria, how likely is it to meet the Magnitude criteria?) In addition to adding a paragraph of discussion, it might be nice to plot all sites as points on the graphs of Figure 3. At least visually, this would allow us to see what the other quadrants look like. I understand that this would be difficult because you are looking at the 10th, 50th and 90th, but I would suggest just showing the 50th in what I describe.

In thinking about the other quadrants of Figure 3, I am curious to hear from the authors about how floods and droughts are changing individually and collectively. Are there many sites where droughts are "hot-spotty" but floods are not? How do these sub-classes compare with your overall conclusions? This, of course, opens other avenues of research beyond the scope of the present work, but I think the authors could provide one or two comments to whet the appetite, so to speak.

Throughout this work I was struck by the unintentional implication that sites not classified as hot-spots will require little changes to water resources management. I think that by showing only positive trends, the implication is that all the other spots (grey dots on figures like Figure 4) are not changing. We all know this is not intended, but I think it might be wise to comment on them. Certainly, the revisions to Figure 3 will help alleviate that unintended implication, but it might also be worth some discussion. As an example, line 3 on page 14 indicates that increases merit changes in management, but I would contend that decreases might also merit changes. I doubt the authors are trying to say that all other spots will be steady as she goes.

LESS GENERAL COMMENTS:

Page 4, line 11: Please cite the package version and the version of R employed.

Page 6, line 10: What constitutes and acceptable number? This seems unnecessarily subjective, so I would like to see the authors provide more objective guidance.

Figure 3: Aside from what was discussed earlier, was there any exploration of significance of these changes? Significance can be somewhat binary, but I am curious if it was considered at all.

Figure 3: The title of (a) should be events per year, right?

Section 3.5: How was seasonality measured? (I'm sorry if I missed it somewhere, but I had trouble finding it.)

Finally, I want to thank you, the authors, for your hard work. It is always a pleasure to review a well-prepared manuscript and I look forward to your continued work.

---

## Author Comment (AC1) · 10 Aug 2018

First of all, we would like to thank you for your interest in our work and your positive and constructive feedbacks. We summarized your comments in the following points and answered them below:

Comment 1: "[. . .] it important to observe that a location might experience a high probability of hydrologic risk without that risk significantly increasing."

Answer to comment 1: We agree with this nuance, there is indeed a difference between a location that presents a high probability of hydrological hazard (and risk), and a location presenting an increasing probability of hydrological hazard. We chose to use the second definition to refer to "hydro-hazard hot-spots", the question behind being:

[Figure]

where should we change our water management practices to better anticipate climate change impacts in terms of hydro-hazards? By doing so, we assume locations where there is a high probability of hydro-hazards are already managed (or at least decision-makers and managers are already aware of these). Our aim is to focus on locations where these risks would intensify or emerge as a result of climate change.

To clarify this, we added a sentence in the introduction: "This is particularly important for regions expected to become even more at risk of both floods and droughts, as these would be 'hot-spots' where resilience to hydro-hazards must be strengthened and water management plans adapted to anticipate climatic changes."

And in section 2.3: "The hot-spot definition aims to clarify the question: 'Where should we anticipate an increase in hydro-hazards as a result of climate change and adapt our water resources management?' By doing so, we assume locations with a high probability of hydro-hazards under the current climate are already managed, known as at risk by decision-makers, and hence do not require highlighting. Instead, our methodology aims to focus on locations where these risks would intensify or emerge in a changing climate."

Comment 2: "[...] locations that experienced some subset of the three criteria described on page 6 and shown in Figure 3. [...] some note about sites that were missed and how strong the correlation between criteria is."

Answer to comment 2: The results we got are so dense that we chose to summarize them in the hot-spot representation for this paper. However, there is much more to say, particularly on sites showing a strong increase in one particular criterion. Hence we decided to add a paragraph in the discussion (section 4.1 page 15) on this topic and add in supplementary information the first maps we produced during this work (Figures S1 and S2), showing changes in each criterion for each hazard and for all the catchments. We agree it would be nice to add points on Figure 3, but tricky to plot all 3 percentiles and we would miss the locations of these points, which is the

additional information we get on the new S1 and S2 maps. They show that correlation between magnitude and duration can be very strong, particularly for droughts, possibly because of how these criteria are calculated (the longer the drought event, the higher the magnitude).

"When analysing changes in each criteria (frequency, magnitude, and duration) separately (Figures S1for floods and S2 for droughts), we can see that for floods the increase in frequency is stronger on the west coast and the southwest of England, while the increase in magnitude is more prominent in the south and southwest of England and duration shows very little changes compared to the other criteria, with the highest increases in the south of England and Wales and the north of Scotland (for the 90th percentile only). For droughts, changes in frequency show a similar spatial distribution (mainly along the west coast), and there is a strong gradient of changes in magnitude (that shows the highest increases compared to the other criteria) and duration, which are strongly correlated, with the highest increases in the diagonal going from the southwest of England up to northeast of Scotland along the west coast of Wales and England."

Comment 3: "[. . .] how floods and droughts are changing individually and collectively."

Answer to comment 3: This is also an interesting point. We have also mapped separated hot-spots for each hazard, keeping the same criteria and increase thresholds (Figure S3), that shows which sites are more "hot-spotty" for one hazard or for the other. This is now described in the discussion (section 4.1).

"Interestingly, when applying the hot-spot analysis separately to each hazard (see Fig. S3), we can see that severe hot-spots (i.e. catchments selected for the 3 percentiles) are shown in southwest of England and Wales and eastern Scotland for floods, and for droughts on the west coast of Wales, England, and Scotland, with 2 catchments on the east coast of Scotland. While there is roughly the same number of hot-spots for both hazards separately, catchments do not necessarily match when floods and droughts

are analyzed together."

Comment 4: "[. . .] the unintentional implication that sites not classified as hot-spots will require little changes to water resources management."

Answer to comment 4: This is a very good point. Since this work focuses on sites where hydro-hazards would worsen, we didn't discuss implications of those where hydro-hazards would decrease, which would of course imply an adjustment in terms of water management planning. A few sentences on this point (locations showing a decreasing in hydro-hazards and possible effects on water resources management) were added in the discussion to nuance our conclusions (section 4.2).

"While this study focused mainly on identifying locations of increasing hydro-hazard hot-spots, Fig. S2 also shows that climatic projections could induce a decreasing drought hazard, particularly in terms of magnitude and duration in the southeast of England and northern Scotland. Such 'positive' changes, i.e. where water deficit would decrease under climate change, would also imply a readjustment of water policies. For example in southeast of England where drought is historically the most frequent observed and managed hydro-hazard, the FFH shows there would be a need to shift hazard management to flood protection, since this region would see an increase in flood frequency, magnitude, and duration, and at the same time a decrease in drought hazard."

Comment 5: Less general comments Answer to comment 5: P4l11: Agreed, that was added in the text. P6l10: Agreed, a more detailed guidance was added on threshold selection in section 2.3 (page 6). Figure 3: We did not investigate significance of changes as such (we did not perform statistical analysis), but severity of changes was implicitly investigated through the 3 percentiles: there is a high severity of change for sites where the 3 percentiles converge (meaning 90% of the ensemble-members agree on the change). Figure 3: Indeed, title of (a) was corrected. Section 3.5: To clarify this point, a description of the mean day of year and seasonality calculation was added in

SI and that was pointed out in section 2.2 (page 4).

Again, thank you for these feedbacks that helped improving this paper.

Please also note the supplement to this comment:
https://www.hydrol-earth-syst-sci-discuss.net/hess-2018-274/hess-2018-274-AC1-
supplement.pdf

―――――――――――――――――
274, 2018.

---

## Referee Comment (RC2) · Anonymous Referee #2 · 21 Aug 2018

Overall Quality: This was a well-prepared manuscript where the researchers presented a novel approach for identifying future floods and droughts in terms of increasing magnitude, frequency, seasonality, as well as overall duration. The research is important as future predictions of floods / droughts have significant meteorological / hydrological implications across various scales of society, especially in a country that has been identified where these extremes are likely to become more common and more management options are needed to address and, potentially, ameliorate these issues.

The researcher's methodology was sound while providing evidence to prove the choice of methodology was not only robust, but scientifically valid. The overall presentation of results are clearly, and concisely stated. Aside from several typographical and grammatical issues, in addition to a few clarifying comments, minor changes are necessary.

[Figure]

Specific Comments:

Page 2 Within the second paragraph, the researchers mention a few examples of uncertainties associated with climate models. It might be best for the researchers to make note that these are only a few examples, and do not constitute the whole of uncertainties surrounded by climate change. The inability to infinitesimally measure climate variables (sensitive dependence on initial conditions) and the concept of chaos theory are two other major factors, for example.

At the end of the second paragraph, are both meteorological and hydrologic drought increasing? Or is only one increasing across the country while the other is largest in the north and west?

At the end of paragraph 3, there is no mention as to the impact of snowmelt. Did the researchers account for snowmelt and its potential impacts on the results? This could have influences based on the results discussed later at the top of page 12.

Perhaps I am not thinking correctly, but what does the fact that Great Britain is an island have anything to do with the results that were achieved? Does this imply you would expect different results from inland countries, or similar results might be seen at Australia? Could potential changing global weather patterns, such as changes in the North Atlantic Drift (which could help to explain why the Western / Southwestern regions of Great Britain see the largest changes in terms of rationality) also be a cause?

Furthermore, could the soil have anything to do with the observed findings?

Not a major issue overall, but something that I found myself struggling with: the color of the figures for time of year may benefit from a slight change (Figures 8 and 9). For

example, the shift from November to December is rather drastic, and the orange-ish red-ish color constantly made me think this represented 'summer'.

After reading through the discussion, were there any indications as to why some of the spatial variability were so extreme? For example, the middle plot-point in Figure 8b shows a mean day of year near May-June, whereas very close next to it, we are back to end of autumn / beginning of winter. Yet, for the 90th percentile (8c), this shifts to be more in-line with the general region (yet there is another example of early-summer surrounding by late-autumn in the northeast of 8c).

The implications are well-discussed, but there was no mention of the potential impacts of urbanization on the hydrology?

Technical Corrections:

Line 9: In the abstract, should this read "Hydrological extremes, floods, and droughts" instead of "hydrological extremes, floods and droughts"?

Line 16: Should there be an apostrophe between 2080 and s (i.e., 2080's)?

Line 21: I would move the fact that this research is novel to the beginning of the abstract to give it more impact early on in the read. Afterwards, expand briefly on why/how it is transferable to new databases, and why that is important? Are not many methods transferrable, or does this add to the validity/impact of the method?

Line 23: See comment about line 16 (page 1). Furthermore, should your first sentence in the abstract and introduction be exactly the same?

Line 24: "UK" has not been previously defined, please make sure you explicitly mention

"United Kingdom" before using the acronym.

Line 28: Change "impact of the recent 2011/12 drought was" to "impact of the drought of 2011/12 was…"

Line 11: Change "with strongest increases in" to "with the strongest increases in"

Line 18: There is an extra period at the end of the sentence / paragraph.

Line 27: Change "dry spells" to "drought periods".

Line 4: Change "However, general floods and droughts are considered independently" to "However, floods and droughts are generally considered independent"

Line 27: See comment about line 16 (page 1). This also appears throughout the rest of the text, please ensure all are accounted for. Or, potentially, since there is a pseudonym for the 1961-1990 period of 'baseline', could there be one for the 2069-2098 period, perhaps simply the 'future'?

Line 5: There is a typographical error associated with the " 50th ".

Line 2: In-text, this citation is from 2012 yet here, it is 2002?

Please also note the supplement to this comment:
https://www.hydrol-earth-syst-sci-discuss.net/hess-2018-274/hess-2018-274-RC2-supplement.pdf
* * *
274, 2018.

---

## Referee Comment (RC3) · Anonymous Referee #3 · 12 Sep 2018

The manuscript "Future hot-spots for hydro-hazards in Great Britain: a probabilistic assessment" provides an innovative approach to identify possible changes in frequency, magnitude, duration as well as the time of both floods and droughts. The paper is in good structure and clear information on results. While I feel some major revisions are required in terms of methodology and discussions before it can be considered further for publication.

1. Introduction: An interpretation on existing approach for projecting future hydro-hazards, not only for UK, but also for other places worldwide, should be included. It will help readers to identify clearly the improvements from the existing knowledge and the innovation of the study. 2. Line 11 Page 4: What is exactly the methodology or indexes of drought characteristics? It is not sufficient to just tell where can find the R

package. 3. Result: It would be much better to show how the climate will change in the scenarios? And for each Figures shown, how the reasons for such changes? Please just be more clear on how these hydro-hazards respond to what kind of changes in which climate parameters?? 4. Line 1 Page 17: Using climate change projections of only one GCM model is only can be acceptable if you show how these characteristics of floods and droughts quantitatively to each unit change in key climate characters. Such information can be also valid for other GCM model outputs.

---

## Author Comment (AC2) · 12 Sep 2018

Thanks you for your positive and constructive feedbacks on our work. Find below the answer to each of your comments:

Specific Comments:

Comment 1: Page 2 – examples of uncertainty sources Answer to comment 1: We agree with this comment, the uncertainty sources we mentioned are indeed not exhaustive, we nuanced the sentence accordingly ("partly due to. . .").

Comment 2: Page 2 – meteorological and hydrological droughts Answer to comment 2: Yes, both of them show these increases, this was clarified in the text ("for both").

[Figure]

Comment 3: Page 4 – snowmelt influence Answer to comment 3: These studies were done in England and Wales, where snowmelt has very little influence, so these authors didn't account for this variable. However the text was modified to reflect the study area of these publications.

Comment 4: Page 11 – the island size influence Answer to comment 4: We see your point. To be clearer: the fact that Great Britain is a relatively small island (compared to Australia for example) implies that space is limited and hence catchments are small, compared to some continental (or large islands like Australia) catchments, and thus river length are short as well as flood duration. This was changed in the text ("relatively small island").

Comment 5: Page 11 – the soil influence Answer to comment 5: Thanks for this interesting question. In a previous paper (accepted in Water Resources Research, to be published soon), we did a regional analysis in Scotland to see which catchment characteristics might influence changes in mean peak flow magnitude (50th percentile in this study). We did not use soil types directly (because it is not part of the catchment characteristics commonly used by flood management consultancies) but rather, the base flow index derived using the Hydrology of Soil Types classification (BFI-HOST), the Standard percentage runoff from the Hydrology of Soil Types classification (SPRHOST), and the proportion of the time that catchment soils are wet (PROPWET), among others. These three characteristics were found to be part of the ones that constrain the spatial distribution of changes in peak flow magnitude the most. Soil types might also have an influence on flood and drought duration. However we did not perform a regional analysis for this study, since it would be out of scope for this paper, which is already very dense and long. But we added a line on this in the discussion, along with the answer to comment 8 on the potential impact of urbanization.

Comment 6: Page 14 – color scale Answer to comment 6: We used on purpose contrasted colors for each season. However, we changed the color scale of the autumn months to highlight the differences between summer and autumn.

[Figure]

Comment 7: Page 15 – extreme spatial variability in mean day of year Answer to comment 7: We need to be cautious in the interpretation of some mean day of year values, particularly for the ones associated to a low seasonality (below 0.6). The example you point out (just like all the values for drought events in Figure 9) is typically of a drought event which mean day of year value (in April) is associated to a low seasonality (below 0.4), and hence is not representative. So for the time of year of drought events, only mean day of year values associated to a medium to high seasonality should be considered, and when looking at these (such as in the 90th percentile in Figure 8), we can see a higher spatial coherence, with drought events in late summer and autumn. What is interesting in this analysis is that the change in seasonality and mean day of year from the baseline to the 2080s: a trend for drought events to become more concentrated in autumn, rather than events spread out though the year.

Comment 8: Page 16 – potential impacts of urbanization Answer to comment 8: We agree with you, urbanization has impacts on drought and flood risks, and have included a sentence within the paper in the discussion to clarify this with a reference which has explored this. However, similarly to the soil influence aspect, this study has not analysed this specifically.

Technical Corrections:

We agreed with these corrections and changed the text accordingly, apart from the "2080s" typography (the native English-speakers of the team use this wording). We also decided to remove the first and last sentences of the abstract which were repetitive to the introduction and did not highlight very much to our work contributions. However, page 5 and 20: we can't see the typographical error associated with "50th", and page 20 the citation is indeed from 2012 (and not 2002), and was appropriately cited in the text and reference.

Please also note the supplement to this comment:
https://www.hydrol-earth-syst-sci-discuss.net/hess-2018-274/hess-2018-274-AC2-

supplement.pdf

[Figure]

**Supplement:**

[revised manuscript text omitted]

---

## Author Comment (AC3) · 20 Sep 2018

Thank you for your interest in our work and your feedbacks. Find below the answer to each of your comments:

Comment 1: Introduction: An interpretation on existing approach for projecting future hydrohazards, not only for UK, but also for other places worldwide, should be included. It will help readers to identify clearly the improvements from the existing knowledge and the innovation of the study.

Answer to comment 1: We agree the introduction is quite UK-centric. We thus added another reference to a European analysis of the impact of climate change on future hydro-hazards by Roudier et al. (2016) who used the euro-CORDEX database.

[Figure]

Comment 2: Line 11 Page 4: What is exactly the methodology or indexes of drought characteristics? It is not sufficient to just tell where can find the R package.

Answer to comment 2: The methodology we used to extract drought characteristics is described throughout section 2.2, particularly Page 4 lines 11-25. To be clearer on which characteristics we assessed, the text was modified line 11 page 4, as well as line 3 page 4 (to detail flood characteristics).

Comment 3: Result: It would be much better to show how the climate will change in the scenarios? And for each Figures shown, how the reasons for such changes? Please just be more clear on how these hydro-hazards respond to what kind of changes in which climate parameters??

Answer to comment 3: We are unclear what you mean by climate parameters: do you mean climatic variables (precipitation, temperature...), or the climatic model parameters? For the later, this can be access through the literature related to HadRM3-PPE [give reference here]. Regarding climate variables, whilst it would be informative to investigate the relationship between the changes in precipitation / temperature/ ... and changes in river flow, it is not the scope of the paper, which aims to identify hydro-hazard hot-spots as a result of climate change (see end of the introduction, we also modified the abstract to clarify the goal). Here, our paper focuses on developing new tools and approaches for water managers to understand potential impact of climate change on water resources, resulting in a relatively dense paper. We believe that the discussion on UKCP09 climatic projections (see section 4.1, page 15 lines 3-13) provides sufficient context in the climate-hydrological modelling chain, and that additional analysis of the climatic variables alone is not necessary.

Comment 4: Line 1 Page 17: Using climate change projections of only one GCM model is only can be acceptable if you show how these characteristics of floods and droughts quantitatively to each unit change in key climate characters. Such information can be also valid for other GCM model outputs.

Answer to comment 4: As mentioned earlier, the aim of this paper to develop a methodology enabling to define probabilistic climate change hydrological impacts. Here it was applied to an ensemble forced by a single GCM, with a perturbed physics parameterization ensemble to encompass some climate modelling uncertainty. The methodology could be easily be applied to a larger ensemble. However, the Future Flows Hydrology ensemble had been analysed in previous work (Prudhomme et al., 2012; section IV) showing that the range and distribution of hydrological changes was generally consistent with the fuller uncertainty described by UKCP09-derived hydrological changes for the 2050s horizon. This was clarified in the discussion (section 4.3). As discussed in this paper, we think a wider range of GCM database would allow a better uncertainty assessment, but the FFH already provides a decent range of possible futures to develop statistical tools.

References: Christel Prudhomme, Sue Crooks, Christopher Jackson, Jon Kelvin, Andy Young (2012) Future flows and Groundwater Levels. Final Technical Report Science Report/Project Note – SC090016/PN9. CEH Wallingford, 118 p.

http://webarchive.nationalarchives.gov.uk/20130301204241/http://www.ceh.ac.uk/sci_programmes/Water/Future%20Flows

Roudier, P., Andersson, J.C.M., Donnelly, C., Feyen, L. Greuell, W., Ludwig, F., 2016. Projections of future floods and hydrological droughts in Europe under a +2°C global warming. Climatic Change, 135: 341. https://doi.org/10.1007/s10584-015-1570-4.

Please also note the supplement to this comment:
https://www.hydrol-earth-syst-sci-discuss.net/hess-2018-274/hess-2018-274-AC3-supplement.pdf

**Supplement:**

[revised manuscript text omitted]